# Coupling Coordination and Dynamic Response Analysis of New-Type Urbanization, Urban Infrastructure and Urban Environment—A Case Study of the Jiaodong Economic Circle

**DOI:** 10.3390/ijerph192214909

**Published:** 2022-11-12

**Authors:** Cheng Lu, Shuang Li, Jiao Liu, Kun Xu

**Affiliations:** 1Safety Science and Emergency Management Research Institute, China University of Mining and Technology, Xuzhou 221100, China; 2School of Economics and Management, China University of Mining and Technology, Xuzhou 221100, China

**Keywords:** NTU-UI-UE, EWM-CCDM-PVAR, the Jiaodong Economic Circle, urban integration

## Abstract

The process of urban development in China is that cities retain independent administrative divisions, realize inter-regional urban integration, and then establish metropolitan areas; this process has high requirements for the coordinated development of cities. China’s cities still need to receive approximately 300 million people in the future, and these urban populations are mainly planned in the emerging metropolitan area. In order to clarify the problems and development of the direction of China’s emerging megalopolis in the process of building a framework for carrying urban population, and to provide reference for China’s urbanization process, the entropy weight method–coupling coordination degree model–panel vector autoregression model (EWM-CCDM-PVAR) is constructed to measure the development level, coupling coordination degree and dynamic response relationship of NTU-UI-UE in the Jiaodong Economic Circle. First, the development level of new-type urbanization and urban infrastructure in the Jiaodong Economic Circle has been steadily improved, but regional differences have been expanding year by year. The urban environment of most cities in the Jiaodong Economic Circle is deteriorating. Second, the CCD level of NTU-UI-UE in the Jiaodong Economic Circle is on the rise, but the CCD level of NTU-UI-UE in Weifang and Yantai is only a Moderate-coupling coordination type, and the development within the region is uneven. Third, both NTU-UI and UI-UE in the Jiaodong Economic Circle have established a dynamic response relationship of mutual promotion, but NTU-UE has a dynamic response relationship of mutual inhibition. Fourth, despite different problems, the Jiaodong Economic Circle has made certain achievements in the process of regional integration and has initially formed a metropolitan circle pattern with Qingdao as its core, Weifang as its west center, Yantai and Weihai as its north center and Rizhao as its south center.

## 1. Introduction

The development process of cities in the world can be seen as the combination of the great growth in the number of cities, the great development of urban circles and the increasingly formed urban agglomerations, as well as the increasingly mature coordinated development pattern of large, medium and small cities and towns, thus, forming a metropolitan circle and moving towards a comprehensive convergence among cities [1]. In the development process, countries across the world have built metropolitan areas by merging and adjusting large-scale cities in recent centuries [2,3]. The basic meaning of urban consolidation refers to the process that urban municipal districts realize scale expansion by absorbing or combining other regions to form a single administrative region [4]. Overall, there are three main types of city consolidation—namely, annexation, consolidation and merger [5]. Annexation refers to the incorporation of unincorporated areas or small towns into the administrative boundaries of a city, and is widely used in developed countries; consolidation refers to the merger of different levels of administrative districts, which is more prominent in the United States, mainly in the form of city–county consolidation [6]; merger refers to the merger of administrative districts at the same level, which is more popular in European countries, mainly between two or more cities [7,8,9,10,11]. Despite the different forms, developed countries are showing a trend of urban expansion and are shaping a new spatial pattern of a “great state and bigger city”. Developed countries have largely completed the urbanization process and established world-class metropolitan areas, while developing countries are entering a period of rapid urbanization, with the BRICS countries of China, Brazil, Russia and South Africa topping the list of urbanization rates and even surpassing some developed countries [12]. In its urban development process, China, the largest developing country, did not follow the process of urban development in developed countries by engaging in large-scale urban mergers, but rather by retaining the separate administrative divisions of cities, while achieving inter-regional urban integration, followed by the establishment of metropolitan areas, with a view to providing the developing world with a sustainable urban development path.

Industrial agglomeration and population concentration are the two twin conditions of urbanization; the concentration of urban population has especially become an important symbol of the development process of urbanization in the world, and is also a necessary condition for the rapid promotion of urbanization and even the formation of metropolitan areas [13]. Globally, the current general urbanization level is approximately 55%, and the United Nations predicts that nearly 1.1 billion new urban residents will be absorbed by cities in 2015–2030 [14]. In China, the most economically dynamic developing country, the process of the integrated development of urban agglomerations to compound metropolitan areas has started the influx of populations to cities, and after 40 years of reform and opening up, China’s urbanization rate has risen from 17.9% in 1978 to 60.6% in 2019 (Figure 1: data from World Development Indicators Database, https://data.worldbank.org.cn, 9 June 2022). The number of the urban population is close to 900 million, which has exceeded the world average urbanization rate by 55%. However, there is still a big gap between the average urbanization rate of developed countries and 80%. China’s “Green Paper on Population and Labor: Report No. 22 on China’s Population and Labor Issues” predicts that China will see a “turning point” of urbanization from high-speed promotion to a gradual slowdown during the “Fourteenth Five Year Plan” period, entering a relatively stable development stage after 2035. The peak rate of China’s urbanization rate is probably 75% to 80%, and China’s cities need to receive approximately 300 million people in the future. In 2019, “The Key Tasks of New-type Urbanization Construction in 2019”, released by China, shows that the urban population in the future is mainly concentrated in more than ten metropolitan areas, especially the emerging metropolitan areas. Under the trend of new-type urbanization in China, new-type urbanization has put forward new requirements for many carriers carrying urban population. As the material carrier of urban population, urban infrastructure needs to build a more comprehensive urban framework for the urban population; for example, the urban environment, as the space carrier of urban population, needs to provide a better living space for the urban population. Therefore, in the process of transformation from traditional urbanization to new-type urbanization, China’s emerging metropolitan area needs to speed up the construction of urban infrastructure and improve the urban environment to accommodate a larger urban population.

The urban system (NTU-UI-UE), composed of new-type urbanization (NTU), urban infrastructure (UI) and urban environment (UE), has been considered to be an important basis for urban population expansion by the academic community. In terms of research themes and contents, there are more studies around the interaction between new-type urbanization, urban infrastructure and urban environment, which mainly involve the establishment of two-dimensional evaluation systems: e.g., new-type urbanization-environment [15], new-type urbanization-infrastructure [16], urbanization-land use [17], economy-environment [18], infrastructure-environment [19], etc. In the research method, the coupling coordination degree (CCD) model [12], dynamic simulation model [17], spatial association model [20] and other methods were used for empirical analysis; alternatively, the development relationship between the two-dimensional systems was analyzed qualitatively in theory [21,22]. Regarding the research area, the current research involved national- [23], province- [24] and prefecture-level cities [25]. At the national level, the study of the NTU-UI-UE coupling coordination relationship at the provincial scale is large, which can easily result in ignoring the spatial and temporal variability of NTU-UI-UE within the province, while the studies at the scale of the prefecture-level city mostly focus on the development within the municipal area, ignoring the integration process between regions. Thus, it is essential to include urban agglomerations in the coupling coordination study of NTU-UI-UE, as most of the urban agglomeration studies are distributed in more mature metropolitan areas. 

Based on the above research, the main deficiencies are as follows: (1) The research scope is insufficient; the above studies adhered to the two-dimensional discussion of new-type urbanization and urban environment, urban infrastructure and population urbanization, new-type urbanization and single infrastructure system, and the multi-dimensional research on new-type urbanization, urban infrastructure and urban environment was lacking. (2) The study area consists mainly of national-, provincial-, prefecture-level cities and mature metropolitan areas, while the future urban population is mainly planned in emerging metropolitan areas; therefore, it is urgent to study emerging metropolitan areas. (3) In terms of research methods, on the one hand, they were used to qualitatively describe the connotation relationship among new-type urbanization, urban infrastructure and urban environment; on the other hand, they were used to quantitatively analyze the coordination relationship among new-type urbanization, urban infrastructure and urban environment based on statistical data, thus, lacking a combination of qualitative and quantitative research methods. Qualitative descriptions are difficult to objectively understand the development trend of new-type urbanization, urban infrastructure and urban environment. Empirical analysis can only explore the interaction between the three and cannot clarify the coordination degree of the three. 

In order to overcome the shortcomings of previous studies, this paper employs the Jiaodong Economic Circle as the research area; constructs NTU-UI-UE; establishes three indicator systems of new-type urbanization, urban infrastructure and urban environment; and measures the development level of new-type urbanization, urban infrastructure and urban environment through the entropy weight method (EWM). NTU-UI-UE forms an organic whole of a city with a complex content structure and complex interaction coupling relationship; the coupling coordination degree model (CCDM) is used to discuss the coupling coordination relationship between new-type urbanization, urban infrastructure and urban environment, and the PVAR model is used to jointly explain the dynamic response relationship between new-type urbanization, urban infrastructure and urban environment in a qualitative and quantitative way.

In other words, this paper established the EWM-CCDM-PVAR model to evaluate the development level, coupling coordination degree and dynamic response relationship of urban NTU-UI-UE; clarify the problems and development direction of China’s emerging metropolitan area in the process of building a framework to support the urban population; and provide a reference for China’s urbanization process, especially in developing countries.

## 2. Study Area and Materials

### 2.1. Study Area

In 2020, the People’s Government of Shandong Province issued Guiding Opinions of Shandong Provincial People’s Government on advancing the integrated development of the Jiaodong Economic Circle; accelerating the integrated development of the five cities of Qingdao, Weifang, Weihai, Yantai and Rizhao in the Jiaodong Economic Circle (Figure 2); building a regional development community with perfect cooperation mechanisms, efficient factor flow, strong development vitality and significant radiation; forming an internationally renowned Jiaodong Economic Circle; and creating a strong engine for high-quality development in the province. This guidance indicated that the Jiaodong Economic Circle should be deeply integrated into the construction of “The Belt and Road”, and should build a new platform for the international cooperation of “The Belt and Road” under the leadership of China-SCO Local Economic And Trade Cooperation Demonstration Zone; the Jiaodong Economic Circle should set up a multimodal transport service system with “one order system” as the core, improve the China–Europe Railway operation platform in Shandong Province and build an international channel connecting Japan and South Korea in the East and Eurasia in the West; the Jiaodong Economic Circle will support Qingdao and Yantai to set up a strategic fulcrum of maritime cooperation, with Weihai constructing China South Korea logistics hub, Weifang building an international power city and Rizhao establishing a Central Asia aviation trade service center. In 2020, the Jiaodong Economic Circle had a population of 32 million and a land area of 52,000 km^2^. The GDP reached 480.4 billion yuan, ranking third among Chinese cities. The urbanization rates of the five cities are all over 60%, and rapid urbanization has brought great pressure upon the carrying capacity of the infrastructure and environment. Therefore, the coordinated development of urbanization, ecological environment and infrastructure has been given more and more attention.

### 2.2. Data Sources and Pre-Processing

This study employed prefecture-level administrative divisions as the basic unit, and analyzed the coupling coordination degree evolution of new-type urbanization, urban infrastructure and urban environment in the Jiaodong Economic Circle from 2010 to 2019. The data were from China Urban Statistical Yearbook, Qingdao Statistical Yearbook, Yantai Statistical Yearbook, Weihai Statistical Yearbook, Weifang Statistical Yearbook and Rizhao Statistical Yearbook from 2011 to 2020. The administrative boundaries were provided by the Resource and Environment Data Cloud Platform, Data Center for Resources and Environmental Sciences, Chinese Academy of Sciences (http://www.resdc.cn/, 6 September 2022). Data for different indexes had different units and magnitudes, making them incommensurable. In addition, different indexes may have had positive or negative effects on the same evaluation objective. To overcome these problems, minimum–maximum normalization was used to process the raw data.

In the process of data collection and processing, the main situations were as follows: (1) The statistical database contained a variety of yearbook sources, there were some inconsistencies in the sources of indexes and the data deviation caused by this statistical situation was relatively small and negligible. (2) The year of individual index data was missing and continuous—it is generally 2–3 years; we adopted the method of linear insertion to improve the data. (3) Because of the data acquisition, the data used were all urban data. In view of the deviation of this kind of formation, this study did not affect the measurement results after eliminating the dimension, therefore, it will not be explained further.

## 3. Research Methodology

The study of the NTU-UI-UE is essentially a study of the interaction relationships between the inner-city systems. Currently, there are studies on the interaction relationships of intra-city systems, most of which use the multi-attribute decision-making (MADM) method for a system-level evaluation, and then introduce CCDM to analyze the interaction relationships between intra-city systems.

Previous studies have successfully applied the MADM method, which has made significant contributions to the evaluation models of new-type urbanization, urban infrastructure and urban environmental level, objective methods such as variance weighting method [26], the fuzzy best–worst method [27], the entropy weight method [12], principal component analysis (PCA) [25] and subjective methods such as G1 [26]. Some studies have also used machine learning algorithms such as random forest for weight calculations and system-level measurements. Random forest has the advantages of effective data processing, indicator weight and correlation determination, high-precision predictions and being insensitive to multicollinearity [28]. A detailed literature review indicates that more applicable evaluation models are needed to help decision makers develop strategies in the NTU-UI-UE evaluation. Although the NTU-UI-UE is a large and complex system, it is characterized by a collection of elements, hierarchy, strong correlations and internal nonlinear interactions. Therefore, it is not possible to view it as a black box model; machine learning algorithms such as random forest based on the black-box theory are not applicable to the study of NTU-UI-UE interrelationships. According to the dissipation theory, the internal positive entropy flow is the main cause of system disorders during the system evolution, and the internal system disorder deepens with the increase in positive entropy. During the evolution of NTU-UI-UE, the interaction of internal system factors as the internal positive entropy flow is the main reason for the disorderly development of the system, so there is a need to clarify the internal system issues, provide materials for the internal system through external urban planning and promote information exchange. Therefore, the entropy weight method, which uses the degree of internal chaos (entropy) of the system as the basis for weight determination, could be selected as the evaluation method to determine the weights of the index factors of NTU-UI-UE.

NTU-UI-UE is a complex system with multiple coupling relationships. For the study of coupling effects, the coupling coordination degree model (CCDM) [12,16], dynamic simulation model [17], spatial association model [20], etc. are commonly used. Among them, the CCDM uses the coupling degree to elucidate the interrelationships among several subsystems, and further uses the CCD to comprehensively evaluate the whole system. Because the model is simple to calculate and the results are intuitive, it has been widely used in empirical studies of the coupling development levels among many systems at different scales and in different regions, such as the environment [29], economy [30], urbanization [15], transportation [31], and population [26]. As far as the intra-city system evaluation is concerned, CCDM is the main method for solving the NTU-UI-UE evaluation, and can clearly illustrate the interactions between the integrated evaluation levels of NTU, UI and UE subsystems.

### 3.1. Theoretical Framework

NTU-UI-UE is a highly complex, multi-layered open system, which is more difficult to coordinate than the coordination of two subsystems, and involves a wider scope. The new-type urbanization and urban infrastructure feed back to each other and develop synergistically: the urban environment provides natural resources and other ecological foundations for the new-type urbanization and urban infrastructure to ensure their positive development, but the rapid development of NTU and UI would have a certain negative impact on the urban environment. Based on previous studies [21,22], we believe that these three subsystems thus form a mechanism of mutual influence and interaction between them (Figure 3).

According to the theory of synergy, the coordinated development of each subsystem is conducive to promoting the orderly and organized communication and cooperation of each element within the whole, thus, achieving the overall function better than the simple sum of the functions of each subsystem; promoting the synergistic development of the three subsystems of new-type urbanization, urban infrastructure, and urban environment; promoting economic circular growth and providing greater welfare for the people in the region. Based on the relationships and synergistic development mechanism, we analyzed the development countermeasures and paths to improve the coordination level of cities by exploring the coupling and coordination characteristics of five cities in the Jiaodong Economic Circle based on four major areas of economic, demographic, social and spatial new-type urbanization; five major systems of energy, water supply and drainage, environment, roads and communication in urban infrastructure; and two major areas of urban ecology—environmental pollution and environmental management. By exploring the coupling coordination characteristics of five cities in the Jiaodong Economic Circle, we finally analyzed the development countermeasures and paths to improve the coordination level of each city.

### 3.2. Index

Based on the actual situation of the Jiaodong Economic Circle, related theories and literature research, we deeply grasped the composition and connotation of new-type urbanization, urban infrastructure and urban environment [32,33]. To ensure the accuracy and feasibility of the evaluation index system, we constructed a scientific, clear, systematic, representative and practical index system according to the following steps: (1).Select indicators, respectively, from the new-type urbanization, urban infrastructure and urban environmental systems.

New-type urbanization focuses on people’s living standards, industrial transformation and upgrading, environmental management, urban planning and innovative social governance [34]. Previous studies mainly analyze the development level of new-type urbanization from five aspects: economic development, social construction, urban population growth, green and sustainable development and urban space expansion [35]. We constructed the environmental infrastructure evaluation indicator system in the urban infrastructure system, therefore, green urbanization is no longer considered specifically. Based on the 4 major domains of economic, demographic, social, and spatial in the new-type urbanization indicators [34,35,36], we measured the level of new-type urbanization construction through 4 primary indicators and 16 basic indicators. First, population urbanization emphasizes population agglomeration, population employment structure and population security, as reflected in previous scholars’ studies [37], therefore, we quantified population urbanization through four indicators: urban population density, urbanization rate, the proportion of employment in the tertiary industry and the number of doctors per 10,000 people. Second, new-type urbanization attaches importance to the transformation of industrial structure and the driving effect on economic development [36]. We selected five indicators (per capita GDP, proportion of the value of the tertiary industry to GDP, total investment in fixed assets, proportion of total imports and exports to GDP and the proportion of scientific expenditure to fiscal expenditure) to quantitatively analyze the level of economic urbanization. Third, the new-type urbanization employs human development as the core goal and aims to improve the quality of life of urban and rural residents and the level of urban public services [15]. Therefore, we selected social urbanization indicators such as the residents’ life, social security, science, education, culture and health. Fourth, spatial urbanization in the context of new-type urbanization is mainly reflected in urban spatial planning and intensive land use [38]. The requirement for urban space is the orderly expansion of urban areas in geographic space, represented by the proportion of built-up areas to urban areas. The inputs and outputs of intensive land use were chosen to be represented by the per-unit area investment in fixed assets and the per-unit area of financial revenue.

We proposed the concept of infrastructure in a broad sense, including engineering infrastructure and social infrastructure. Since both social infrastructure and social urbanization reflect the degree of social construction in cities, we established a social urbanization indicator system in the new-type urbanization system so that social infrastructure was no longer involved in the infrastructure system. Because urban infrastructure indicators can describe the function of urban infrastructure—providing basic functions for urban residents and playing an important role in supporting urban socio-economic activities—and the main body of the role is the urban population, the per capita and ratio indicators were chosen to better reflect the level of urban infrastructure supply. To measure the level of urban infrastructure supply set up, we selected 5 primary indicators and 14 secondary indicators based on 5 major systems of urban infrastructure indicators: energy, water supply and drainage, environment, roads and communication [39,40,41,42,43]. Among the energy systems, the gas penetration rate is more representative of the gas supply level compared to gas supply, therefore, the indicator of annual electricity consumption per capita was chosen to reflect the basic domestic electricity supply [39] compared to electricity consumption. The indicators of the per capita daily consumption of tap water for residential use and the coverage rate of the urban population with access to tap water were selected to analyze the supply level of the water supply facility system; the density of a drainage pipeline in a developed area and the rate of sewerage disposal can visually reflect the urban drainage and water treatment capacity [40]. In the transportation facility system, the number of taxis and buses per 10,000 people was selected to reflect the basic transportation services in the city, and the per capita area of roads was chosen to reflect the convenience of urban transportation [42]. Only Qingdao had opened an urban rail transit, therefore, the urban rail transit mileage indicator was deleted. Mobile phone and Internet penetration rates were selected to quantify and analyze the level of communication network construction in cities [43]. Urban green space has the functions of improving urban livability, increasing urban aesthetics, purifying air and preventing and mitigating disasters, and is the basis of the urban environmental facility system; therefore, two indicators of the per capita park green areas and green coverage rate of developed areas were chosen to measure the construction of urban green space. Public toilets can improve the environmental treatment capacity of cities and are a necessary infrastructure; therefore, public toilets per 10,000 population were chosen to reflect the image construction of urban environmental infrastructure.

We used environmental pollution and environmental governance to describe the urban environment [44,45]. Environmental pollution and environmental management mainly refer to the pollution and management of air, water and soil resources; therefore, we chose a proportion of heavily polluted weather and SO2, industrial solid waste and a dust (soot) emission volume to represent air pollution; wastewater discharge to represent water pollution; chemical oxygen demand, NHx and the NOx emission volume to represent soil pollution; per capita water resources and the rate of sewerage disposal to reflect the water environment management; the green coverage rate of developed areas to reflect the soil environment management; and the comprehensive utilization rate of industrial solid waste to reflect air pollution management.

(2).Collinearity diagnostics of the index system: considering the collinearity between the selected indexes [46], the collinearity diagnosis of the evaluation indexes of new type urbanization, urban infrastructure and urban environment was carried out, respectively. After the above steps, the results showed that the collinearity diagnostics of each index met the research requirements.(3).Construction of index system: to determine the evaluation index system. The final evaluation index system is shown in Table 1. Index is divided into three levels in Table 1, the first-level index is the system layer, the second-level index is the subsystem layer and the third-level index is the index layer; Nj, Ij, Ej represent the third-level index of NTU, UI, UE, respectively, + indicates a positive index, − indicates a negative index.

### 3.3. Evaluation of the New-Type Urbanization and Urban Infrastructure, Urban Environment (NTU-UI-UE) Subsystems

NTU-UI-UE is a huge complex system which has the characteristics of an element set, clear hierarchy, strong correlation and internal nonlinear effect. According to the dissipation theory, the internal positive entropy flow is the main cause of system disorders in the process of system evolution. With the increase in positive entropy, the degree of the internal system disorder is deepened [47].

In the evolution process of NTU-UI-UE, the interaction of internal factors of the system as the internal positive entropy flow is the main reason for the disordered development of the system. Therefore, it is necessary to clarify the internal problems of the system, provide materials for the internal system through external urban planning, and promote information exchange. Therefore, to determine the weight of NTU-UI-UE indicators, the entropy weight method based on the degree of chaos (entropy) within the system should be selected as the evaluation method.

Entropy is an important concept in the second law of thermodynamics to characterize the state of matter, which was originally introduced to the information theory by Shannon [48] to represent the degree of uncertainty, and is now commonly used in the field of urban development [12]. 

According to the methods above, the collected data were standardized by Formula (1) in positive and negative dimensions, respectively.
(1)$$\left\{\begin{array}{c} Y_{ij}^{+}=\frac{X_{ij}-X_{\min}}{X_{\max}-X_{\min}}\\ Y_{ij}^{-}=\frac{X_{\max}-X_{ij}}{X_{\max}-X_{\min}} \end{array}\right.$$
where $Y_{ij}$ represented the value of index j, in year *i*; i=1, 2,…,n; j=1, 2,…,m;

$X_{\max}$ and $X_{\min}$ were, respectively, the maximum and minimum value of index j.

$Y_{ij}\in \left[0,\;1\right]$, as the number of indicators grew, the subsystem became better, which indicates that the influence of the indicators on the subsystem was positive. Conversely, the influence of the indicators was negative.

First, we calculated the proportion of the sample index $p_{ij}$ by: $p_{ij}=Y_{ij}/\sum_{i=1}^{n}Y_{ij}$, and in order to avoid the case of $\ln(p_{ij})=0$, the denominator and numerator of $p_{ij}$ needed to pulse the number 0.1, by Formula (2):(2)$$p_{ij}=(0.1+Y_{ij})/\sum_{i=1}^{m}\left(0.1+Y_{ij}\right)$$

Second, we calculated the entropy of index j by Formula (3):
(3)$$e_{j}=-\frac{1}{\ln m}\sum_{i=1}^{n}p_{ij}\ln p_{ij}$$

The informational entropy weight $W_{j}$ can be calculated by Formula (4):(4)$$W_{j}=(1-e_{j})/\sum_{i=1}^{m}\left(1-e_{j}\right)$$

The final weights are shown in Table 1.

Third, we calculated the comprehensive index $U_{i}$ as Formula (5):(5)$$U_{i}=\sum_{j=1}^{m}W_{j}Y_{ij}^{+}(or\;Y_{ij}^{-})$$

### 3.4. The Coupling Coordination Degree Model (CCDM) of New-Type Urbanization, Urban Infrastructure and Urban Environment Subsystems

The coupling degree C is the core part of CCDM, and the value of C should be in [0, 1] to indicate the strength of the coupling relationship between systems. The wrong use of CCDM mainly comes from the wrong coupling degree C formula, which leads to the C result interval not being [0, 1], thus, causing a series of errors in the results and interpretation of the subsequent coupling coordination degree. The CCDM of multiple subsystems are mainly divided into two categories. The first category is Equation (6), which has been used by Ma [49] and Liu [50] to calculate the C value. Jiang [51] proved that the value range of $C_{1}$ calculated by Equation (6) is $\left[0,\frac{1}{2}\right]$, thus, Equation (6) will underestimate the coupling degree; therefore, the conclusion obtained by using Equation (6) to calculate the C value is not valid. The second category is Equation (7), which has been used by Ge and other scholars [52] to calculate the value. Jiang [51] and Wang [53] proved that the value range of coupling degree $C_{2}$ in Equation (6) is between [0, 1]; when, and only when, $U_{1}=U_{2}=\cdots U_{n}$, the coupling degree reaches the maximum value of 1. Equation (7) is correct.
(6)$$C_{1}={\left[\frac{U_{1}\times U_{2}\times \cdots \times U_{n}}{\prod_{i\neq j}\left(U_{i}+U_{j}\right)}\right]}^{\frac{1}{n}}$$
(7)$$C_{2}=\frac{{n\left[U_{1}\times U_{2}\times \cdots \times U_{n}\right]}^{1/n}}{U_{1}+U_{2}+\cdots +U_{n}}$$

Therefore, this paper uses Equation (7) of Ge [52] to calculate the coupling value.

Some scholars use the geometric weighting method to calculate the overall development level T value of the system. Jiang [51] verified that the value range of the geometric weighting method is $\left[0,\frac{1}{2}\right]$, and the overall development level is also underestimated. Therefore, this paper uses the arithmetic weighting method to calculate the value. See Equation (8) for details.
(8)$$T=\alpha_{1}U_{1}+\alpha_{2}U_{2}+\cdots +\alpha_{n}U_{n}$$
where, $U_{1},U_{2},\cdots ,U_{n}$ are the comprehensive development levels of each subsystem, and the values of $U_{1},U_{2},\cdots ,U_{n}$ are determined by Equation (5). This paper assumes that each subsystem is equally important to the coordinated development of the NTU-UI-UE, so $\alpha_{1}=\alpha_{2}=\cdots =\alpha_{n}=\frac{1}{n}$.

The CCDM is built based on the coupling degree and system development level, and its equation is as follows:(9)$$D=\sqrt{C\cdot T}$$

### 3.5. Classification Criteria for the CCD

The main purpose of delineating the type of coupling coordination degree is to determine whether the NTU-UI-UE is healthy or not, and to clarify the stage of coordinated development of the system in order to provide suitable solutions. By referring to the research of Jiang [51] and Xing [30], we classified the coupling coordination degree into four broad categories: disorderly development, fragile development, coordinated development and benign development. Specifically, when D≤0.25, the system is in a disorderly development state, the development level of NTU, UI and UE systems are all low, and the interaction relationship between the three is weak; when $D\in \left(0.25,0.5\right)$, the system begins to develop in a coordinated but fragile way, the new-type urbanization and urban infrastructure are in the early stage of rapid development, the development level is low but the development speed is fast—however, at the expense of part of the urban environment—and the urban environment development is unstable. When $D\in \left(0.5,0.75\right)$, the system coordinated development, new-type urbanization and urban infrastructure development levels are high but slowing down, and the good urban framework built by new-type urbanization and urban infrastructure starts to improve the environment. When $D\in \left(0.75,1\right)$, the system belongs to the benign coupling stage; the new-type urbanization construction, urban infrastructure supply and urban environment development levels are high; and the NTU-UI-UE internal development complements and promotes each other. This stage of the system follows the principle of local subordination for the whole, dynamic development; it has openness to the outside and is able to adapt and change the environment to ensure the overall stable upward development of the system. D always satisfies $D\in \left(0,1\right)$; when D=0, the system is in a state of collapse, the level of new-type urbanization construction and urban infrastructure supply are extremely low, and the urban environment is extremely poor. When D=1, the system moves to a new orderly structure—the development level of NTU, UI and UE reaches the highest level, and the coupling coordination degree reaches the optimal level. Further, we refined the coupling coordination degree into eight subcategories based on the above four classification divisions; the specific classifications and explanations are shown in Table 2.

### 3.6. PVAR

This paper adopts the Panel Vector Autoregression (PVAR) model proposed by Love and Zicchino; the PVAR model has several econometric advantages to make it the best method to test macroeconomic dynamics [54]. First, PVAR is helpful to analyze the impact propagation between variables in unit time. Second, PVAR is based on the analysis of real data series, rather than adhering to the concept of macroeconomic. Third, the model does not lead to differences between dependent variables and independent variables, but regards all factors as mutually endogenous. In addition, it also provides the interactive response process of dependent variables and independent variables. The PVAR model is shown in Equation (10):(10)$$y_{it}=\beta_{0}+\sum_{j=1}^{k}y_{i,t-j}+\gamma_{i}+\delta_{t}+\varepsilon_{it}$$

In Equation (10), $y_{it}$ is a column vector containing 3 variables; i and t represent provinces and time, respectively; $\beta_{0}$ represents intercept items; $\beta_{j}$ represents the coefficient matrix of lag order *j*; $\gamma_{i}$ represents regional fixed effect; $\delta_{t}$ represents the time-fixed effect and $\varepsilon_{it}$ represents the stochastic error. 

## 4. Results

### 4.1. Comprehensive Development Levels

The comprehensive development trend of the new-type urbanization system of the NTU-UI-UE is shown in Figure 4. During 2010–2019, the development level of NTU in the five cities of the Jiaodong Economic Circle has improved and shown relatively stable growth. The level of NTU in Qingdao has always been the first. From 2014 to 2016, the development of NTU in Qingdao slowed down, and then rebounded strongly in 2017–2019. The level of NTU in Weihai has declined in the two cycles of 2010–2012 and 2016–2017, and its development has stagnated. Weihai has always maintained the advantage of the second level of NTU in the Jiaodong Economic Circle, but the development speed of NTU is relatively slow, and the gap between Yantai and Weihai is slowly narrowing. The level of NTU in Weifang and Rizhao is relatively backward, and the gap with Qingdao, Yantai, Weihai tends to expand. From the perspective of the growth rate of NTU development, Weifang, Rizhao and Yantai have maintained a stable growth rate, Qingdao has the fastest growth rate and Weihai has the lowest growth rate. In general, the level of the NTU in the Jiaodong Economic Circle has shown a steady upward trend, rising from 0.23 in 2010 to 0.59 in 2019. However, the NTU level gap among cities in the Jiaodong Economic Circle has been further widened.

Figure 5 shows the trend of the level of urban infrastructure development in NTU-UI-UE. During 2010–2019, the UI level of the five cities in the Jiaodong Economic Circle has maintained a steady upward trend. The UI level of Weihai is growing rapidly. From 2011 to 2019, Weihai has always been the city with the highest UI construction level in the Jiaodong Economic Circle. The UI level of Qingdao remains the second highest in the the Jiaodong Economic Circle, the growth rate is unstable and has declined in several development cycles. Yantai has the slowest growth rate of UI level, with a significant decline in 2013–2014 and a “0” growth in 2015–2018. The UI level of Rizhao and Weifang has kept relatively stable improvement. In the past ten years, the UI level of Rizhao and Weifang has both increased to 0.60, but they are still at the bottom of the five cities, and the gap between cities is widening. In general, the UI level of Jiaodong Economic Circle cities shows a significant upward trend, rising from 0.32 in 2010 to 0.67 in 2017. However, the gap of UI level among cities in the Jiaodong Economic Circle has been further widened.

Figure 6 shows the development trend of the UE level in the Jiaodong Economic Circle. From 2010 to 2019, the UE level of Qingdao, Yantai, Weihai and Rizhao all showed a downward trend to varying degrees, Weifang was the only exception to this trend, and the UE level of Weifang City increased from 0.25 to 0.52. During the study period, the UE level in Yantai was the lowest, while that in Weihai was the highest. In general, the UE level of the Jiaodong Economic Circle is very unstable, and the UE level remains at 0.62.

### 4.2. Coupling Coordination Degree Results

From 2010 to 2019, the CCD development of NTU-UI-UE in the five cities of the Jiaodong Economic Circle show a very stable upward trend (Figure 7). The CCD development of NTU-UI-UE in the Jiaodong Economic Circle can be divided into two stages: the initial development stage of 2010-2014, where there is no policy guidance. In March 2014, the State Council issued “the National new-type urbanization Plan (2014–2020)”, which marks that Chinese cities have officially entered the era of new-type urbanization. In the stable growth stage of 2015–2019, after the promulgation of the new-type urbanization plan, the Jiaodong Economic Circle has policy guidance. Although the five cities have different NTU, UI and UE system problems, the Jiaodong Economic Circle has entered a new development stage.

The initial development stage of 2010–2014. In 2010, the CCD level of NTU-UI-UE in Qingdao and Weihai was the modern coupling coordination type, that of NTU-UI-UE in Yantai and Rizhao was the primary coupling coordination type, and that of NTU-UI-UE in Weifang was the mill discharge type. After five years of initial development, in 2014, the CCD level of NTU-UI-UE in Weihai was the favorable coupling coordination type; the CCD levels of Qingdao, Yantai and Rizhao NTU-UI-UE were all improved to the modern coupling coordination type; and the CCD level of Weifang NTU-UI-UE was developed to the primary coupling coordination type. In the initial development process, the CCD level of the Jiaodong Economic Circle NTU-UI-UE maintained a steady growth.

In the 2015–2019 development stage, the CCD level of NTU-UI-UE in five cities was still growing steadily. The CCD level of NTU-UI-UE in Qingdao, Weihai and Rizhao was the first to become the favorable coupling coordination type. The CCD level of NTU-UI-UE in Yantai and Weifang was the modern coupling coordination type. In the later stage of development, under the policy guidance of “the National new-type urbanization Plan (2014–2020)”, it was required to form a new-type urbanization pattern with urban agglomeration as the main body and coordinated development of large-, small- and medium-sized cities and towns. The CCD level of NTU-UI-UE in the Jiaodong Economic Circle continued to improve steadily, with an average growth rate higher than that of the previous cycle.

In general, the internal gap of the Jiaodong Economic Circle did not narrow, and the CCD level of the NTU-UI-UE in Weifang and Yantai was still relatively backward, which was only the modern coupling coordination type. The UE subsystem of the entire Jiaodong Economic Circle had major problems.

### 4.3. PVAR Results

This paper mainly uses Eviews11 for data processing and analysis. First, the panel data NTU level, UI level and UE level are logarithmically processed. After logarithmic processing, the variables NTU, UI and UE are named LNNTU, LNUI and LNUE, respectively. Then, the unit root test (PP-Fisher Chi square test) was carried out for variables. The p value of the unit root test for variables LnNTU, LnUI and LNUE were all less than 0.05, and the original hypothesis was rejected. The data were stable, and the PVAR model could be built.

Before building the PVAR model, the optimal lag order of the model should be determined. According to LR, FPE, AIC, SC and HQ criteria, the optimal lag order was determined as order 1 (Table 3).

The purpose of the robustness test is to ensure that the model is effective. The points in the unit circle in Figure 8 represent the inverse roots of the AR polynomial characteristic. If these points fall in the unit circle, the model is stable. It can be seen from Figure 1 that the PVAR model is robust.

The impulse response function is used to analyze the response of a variable to the dynamic impact of other variables. The sample data structure lags behind by 10 periods. The results are shown in Figure 9, Figure 10 and Figure 11. The specific analysis leads to the following conclusions:(1).Figure 9 is the LNNTU pulse response diagram. When LNNTU is impacted by itself, it forms a long-term positive effect, reaches the peak in the first period, and then slowly weakens. The impact of LNUI on LNNTU showed a trend of first rising and then stabilizing. From the initial stage to the third stage, the promotion effect was enhanced, and from the fourth stage to the tenth stage, it tended to be stable. LNUI maintained a positive promotion effect on LNNTU. LNUE had a relatively stable inhibitory effect on LNNTU from the second period to the tenth period. To sum up, LNNTU is promoted by itself and LNUI, but LNUE inhibits LNNTU.(2).Figure 10 is the pulse response diagram of LNUI. When LNUI is impacted by itself, there is a long-term positive effect, which reaches the highest at the initial period, and then decreases steadily. The impact of LNNTU on LNUI showed a weak negative effect in the first period and maintained a relatively stable positive effect from the second period to the tenth period. The positive promotion effect of LNUE on the stability of LNUI was maintained from the first period to the tenth period. The above relationship shows that the main driving force of LNUI comes from itself, NTU and UE, among which, LNNTU plays a weak role in promoting LNUI.(3).LNUE has formed a long-term positive effect when it is impacted by itself, reaching the highest impact in the first period, and then weakening steadily. LNNTU has a long-term weak negative inhibition on LNUE. From the second period to the eighth period, LNUI has a long-term positive role in promoting LNUE. This shows that the main driving force of UE also comes from the dual drive of itself and UI, and NTU inhibits UE. The above relationship shows that the main driving force of LNUE comes from itself and LNUI, but LNNTU inhibits UE.

To sum up, LNNTU is promoted by itself and LNUI, and LNUE inhibits LNNTU. LNUI is promoted by itself, LNNTU and LNUE, and the promotion of LNNTU on LNUI is weak. LNUE is promoted by itself and LNUI, and LNNTU inhibits LNUE. NTU, UI and UE have a strong self-promotion effect, which indicates that the development of NTU, UI and UE has a large “inertia”. NTU and UI promote each other, which indicates that NTU and UI have established a good dynamic response relationship. UI and UE promote each other, which indicates that UI and UE have also established a good dynamic response relationship. NTU and UE inhibit each other, which indicates that NTU and UE have not established a good dynamic response relationship.

## 5. Discussion

The development level of new-type urbanization and urban infrastructure in the Jiaodong Economic Circle has been steadily improved; however, there are regional differences in the development of new-type urbanization and urban infrastructure in the Jiaodong Economic Circle, and the differences are expanding year by year. The urban environment development fluctuates greatly, and most cities in the Jiaodong Economic Circle show a downward trend. The CCD level of NTU-UI-UE in the Jiaodong Economic Circle is on the rise, but the CCD level of NTU-UI-UE in Weifang and Yantai is only a moderate coupling coordination type, so the internal development of the Jiaodong Economic Circle is uneven. New-type urbanization and urban infrastructure in the Jiaodong Economic Circle have established a dynamic response relationship of mutual promotion, and the urban infrastructure and urban environment have a dynamic response relationship of mutual promotion, but new-type urbanization and urban environment show a malignant relationship of mutual inhibition.

From 2010 to 2019, the new-type urbanization of the Jiaodong Economic Circle inhibited the development of urban environment, and the urban environment was extremely unstable. At present, China has basically met the economic driving conditions to cross the inflection point of EKC, but the lagging urbanization process will lead to the flattening and fluctuation of the peak pollutant emissions. This is mainly because the rapid development of urbanization has a certain rigid demand for industrialization, especially the development of heavy industry. Although many factors that promote the reduction in environmental pollution have already existed, the offset of a rigid demand will lead to environmental instability. The above discussion was verified in Wang’s research [55]. Although there are policies to reduce urban environmental pollution in the new-type urbanization plan, the industrial structure with a large proportion of heavy industries in Jiaodong Economic Circle has led to the instability of the urban environment. This is the reason why new-type urbanization inhibits the urban environment, and it is also the reason why the urban environment development is unstable.

From 2010 to 2019, the urban environment of the Jiaodong Economic Circle has always inhibited the development of new-type urbanization. This paper points out that urban environment includes urban greening, urban pollution, environmental governance and natural environmental resources. Problems of these urban environmental factors will lead to poor living conditions of the urban population, which is not conducive to the process of urbanization. This is consistent with Xia’s research conclusion—Xia shows that urban environmental pollution will hinder the process of urbanization and will directly hinder the development of cities into “global metropolises” [56].

Therefore, it is reasonable that new-type urbanization and urban environment have a mutually restrictive relationship; Zou et al. [57] also reached the same conclusion.

From 2010 to 2019, the level gap between new urbanization and urban infrastructure in the Jiaodong Economic Circle has been further expanded, and regional differences in the coupling coordination development still exist, which is very unfavorable to the construction of a metropolitan area. The path of urban development in China is not characterized by large-scale urban mergers, but by the dynamic formulation of macro-policies to regulate urban development according to the stage of urban development. The development of new-type urbanization and urban infrastructure is largely influenced and affected by national macro-regulation, policy, institutional reform and local government actions. The macro development policies of the cities in the Jiaodong Economic Circle are consistent, but there are great differences in the population, economic aggregate, urban scope and consumption level among the cities in the Jiaodong Economic Circle, and problems of new-type urbanization and urban infrastructure vary from city to city. In the case of consistent policies, the differences in urban conditions make it difficult for cities such as Rizhao and Weifang with poor urban foundations to make up for the differences.

The development of new-type urbanization and urban infrastructure is largely positively influenced and affected by national macro-regulation, policy, institutional reform and local government actions, but the urban environment does not improve immediately under the macro-regulation and policy-driven effects, and the development process of new-type urbanization and urban infrastructure will definitely require the urban environment to be sacrificed. However, both will also improve the urban environment after their development reaches a certain level. Therefore, in the process of urban development, the coordinated development of NTU-UI-UE is not stable, and controlling the floating range rather than blindly demanding urban environmental quality can ensure the sustainable development of the whole system, which has important implications for city managers to formulate urban planning policies. The development of most Chinese cities has not yet reached the inflection point of EKC. Therefore, the coupling coordination development of NTU-UI-UE is essentially a sustainable urban development path that can promote the overall downward movement of EKC and accelerate the emergence of the inflection point of EKC. The integrated construction between regions depends on the coordinated development within the regions on the one hand, and the synergistic effect between regions depends largely on the complementary industrial structure between regions on the other hand. When the coupling coordination development of NTU-UI-UE reaches a certain level, the degree of complementary industrial structure should be a further urban issue for city managers to think about.

## 6. Conclusions

Based on the panel data of five cities in the Jiaodong Economic Circle from 2010 to 2019, this paper establishes the EWM-CCDM-PVAR model to analyze the development level, coupling coordination degree and dynamic response relationship of NTU-UI-UE in the Jiaodong Economic Circle. The results show that: (1) The development level of new-type urbanization and urban infrastructure in the Jiaodong Economic Circle has been steadily improved, but there are regional differences in the development of new-type urbanization and urban infrastructure in the Jiaodong Economic Circle, and the differences are expanding year by year. The overall development is uneven. The development of the urban environment fluctuates greatly, and the urban environment of most cities in the Jiaodong Economic Circle is deteriorating. (2) The CCD level of NTU-UI-UE in the Jiaodong Economic Circle is on the rise, but the CCD level of NTU-UI-UE in Weifang and Yantai is only a moderate coupling coordination type, and the development within the region is uneven. (3) NTU and UI of the Jiaodong Economic Circle have established a mutually reinforcing dynamic response relationship, and UI and UE have also established a mutually reinforcing dynamic response relationship, but NTU and UE have a mutually inhibiting dynamic response relationship. (4) During the ten-year development process of establishing a metropolitan area by retaining the independent administrative divisions of cities and achieving interregional urban integration in the Jiaodong Economic Circle, the development levels of new-type urbanization, urban infrastructure and the coupled coordination degree level of the NTU-UI-UE system have been substantially improved, and the regional integration continues to deepen. Under the coordinated governance and mutual achievements of new-type urbanization, urban infrastructure and urban environment, the Jiaodong Economic Circle has formed a metropolitan circle pattern with “one core” and “four wings”, which confirms that the urban development path with Chinese characteristics is feasible and correct.

## 7. Implications for Urban Integration

We put forward the following suggestions for the urban development of the new economic circle: First, the urbanization process will bring environmental pollution, and the poor urban environment will also hinder the urbanization process. Different environmental governance measures should be implemented according to different industrial structures in different regions. There are many heavy industries in the Jiaodong Economic Circle. Environmental policies and measures aimed at strengthening the end treatment and clean transformation of high-polluting industries and improving energy efficiency can help the Jiaodong Economic Circle quickly surpass the peak of pollutants. Second, in the process of urban integration in the emerging economic circle, regional differences are the main problem. The industrial structure is the fundamental problem that hinders urban integration. In the process of the integration of the emerging metropolitan area, it is necessary to promote the industrial complementation of the cities within the metropolitan area. World class metropolitan areas have formed distinctive industrial clusters [58]. Drawing on the experience of world class metropolitan areas, the Jiaodong Economic Circle should coordinate the regional industrial division and cooperation system. Weifang and Yantai have obvious advantages in the manufacturing industry; Yantai and Rizhao have obvious advantages in the tourism service industry; Qingdao has developed finance, trade and logistics—the five cities have the basis of industrial complementation, and the prospect of industrial cooperation is great. A higher level of industrial cooperation will make up for regional differences.

## Figures and Tables

**Figure 1 ijerph-19-14909-f001:**
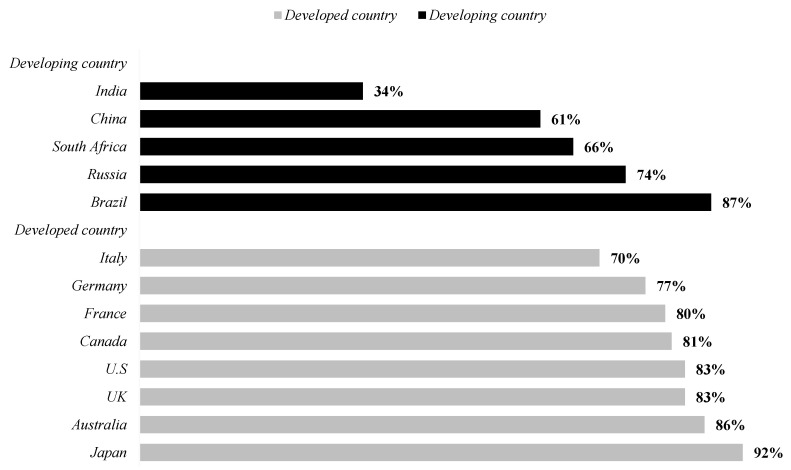
The stage of urbanization in some countries of the world in 2019.

**Figure 2 ijerph-19-14909-f002:**
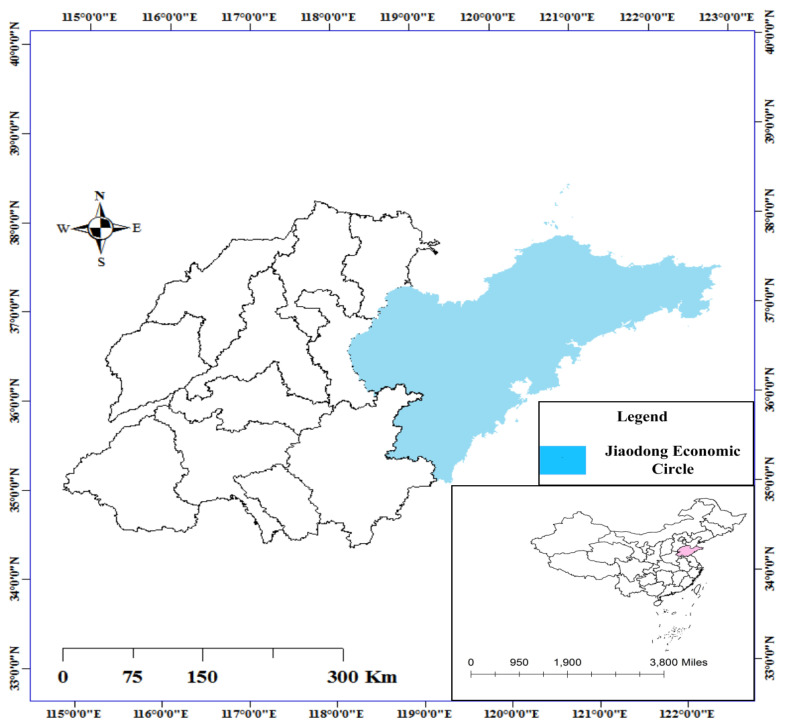
Study area.

**Figure 3 ijerph-19-14909-f003:**
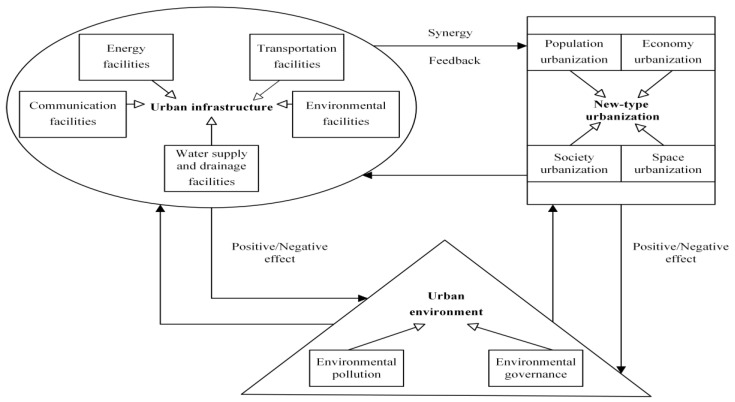
Relationship network of new-type urbanization, urban infrastructure and urban environment (NTU-UI-UE).

**Figure 4 ijerph-19-14909-f004:**
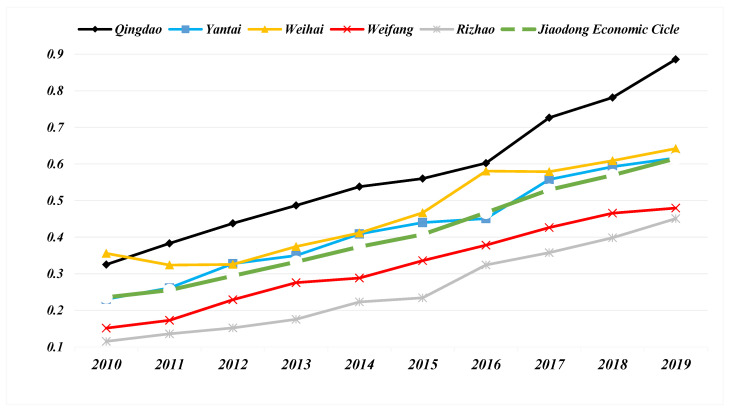
Timeseries of the comprehensive new-type urbanization level of 5 cities and the entire Jiaodong Economic Circle.

**Figure 5 ijerph-19-14909-f005:**
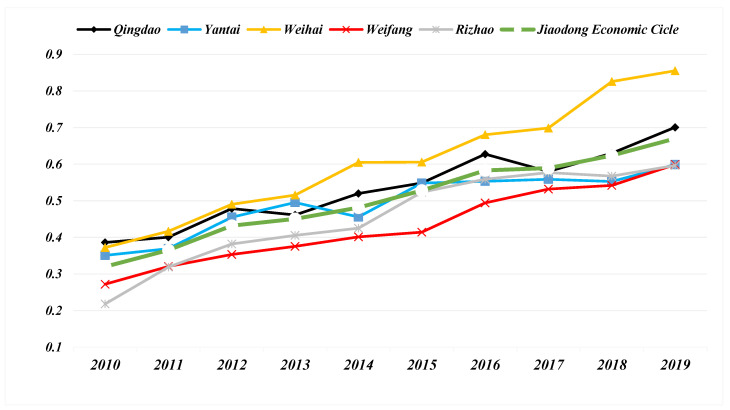
Timeseries of the comprehensive Urban infrastructure level of 5 cities and the entire Jiaodong Economic Circle.

**Figure 6 ijerph-19-14909-f006:**
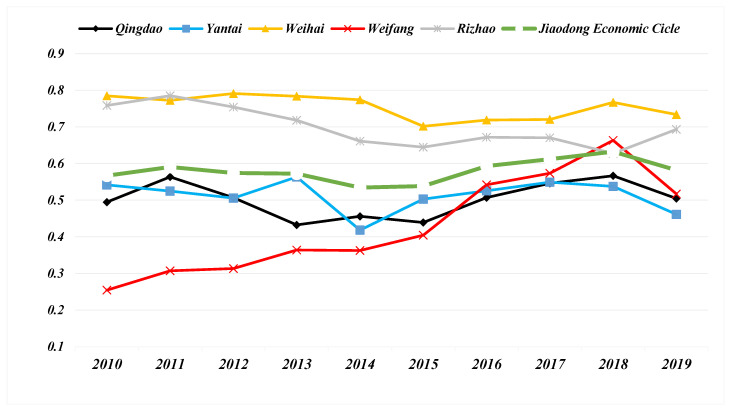
Timeseries of the comprehensive Urban environment level of 5 cities and the entire Jiaodong Economic Circle.

**Figure 7 ijerph-19-14909-f007:**
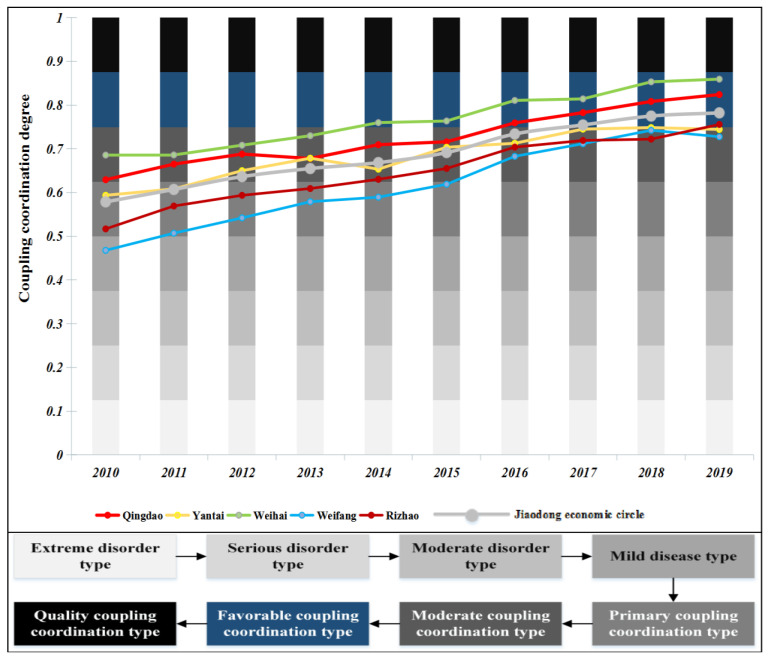
Timeseries of the coupling coordination degree of 5 cities and the entire Jiaodong Economic Circle.

**Figure 8 ijerph-19-14909-f008:**
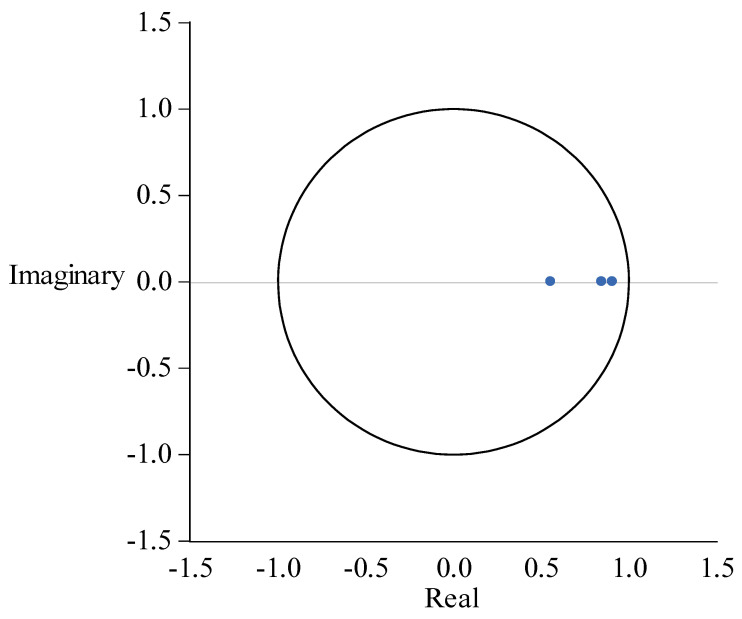
PVAR model robustness test result.

**Figure 9 ijerph-19-14909-f009:**
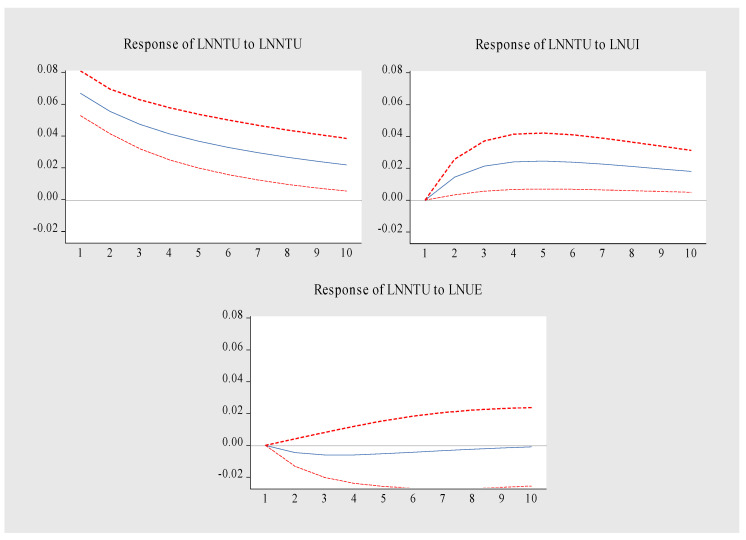
Pulse response diagram of LNNTU.

**Figure 10 ijerph-19-14909-f010:**
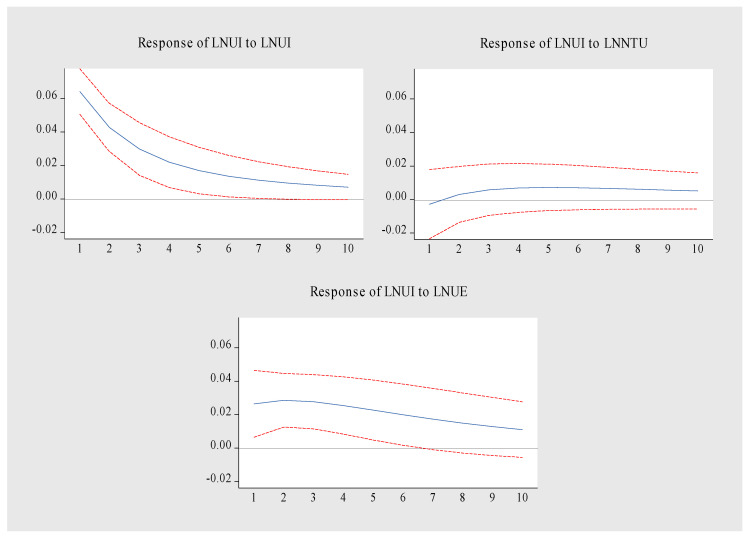
Pulse response diagram of LNUI.

**Figure 11 ijerph-19-14909-f011:**
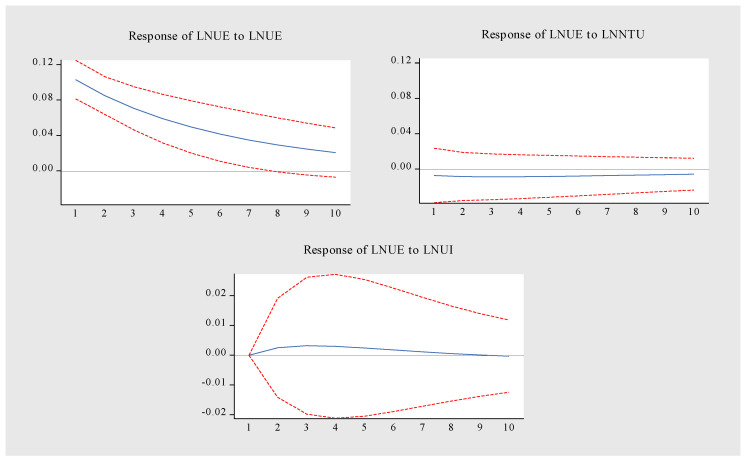
Pulse response diagram of LNUE.

**Table 1 ijerph-19-14909-t001:** New-type urbanization, urban infrastructure and urban environment index system and weight.

System Layer	Subsystem Layer	Index Number	Index Layer (Unit)	Attribute	Weight
New-type urbanization	Population urbanization	N1	Urbanization rate (%)	+	0.071
		N2	Urban population density (people/km^2^)	−	0.074
		N3	Proportion of the employment in tertiary industry (%)	+	0.060
		N4	Number of doctors per 10,000 people (people)	+	0.040
	Economy urbanization	N5	Per capita GDP (Yuan)	+	0.061
		N6	Proportion of the value of the tertiary industry to GDP (%)	+	0.069
		N7	Total investment in fixed assets (Yuan)	+	0.090
		N8	Proportion of total imports and exports to GDP (%)	+	0.042
		N9	The proportion of scientific expenditure to fiscal expenditure (%)	−	0.039
	Society urbanization	N10	Urban per capita disposable income (Yuan)	+	0.055
		N11	Number of beds in medical institutions per 10,000 population (unit)	+	0.047
		N12	Proportion of social security and employment expenditure to financial expenditure (%)	+	0.038
		N13	Engel’s coefficient (%)	−	0.059
	Space urbanization	N14	Proportion of built-up areas to urban areas (%)	+	0.080
		N15	Per-unit area financial revenue (Yuan/km^2^)	+	0.086
		N16	Per-unit area investment in fixed assets (Yuan/km^2^)	+	0.090
Urban infrastructure	Energy facilities	I1	Gas penetration rate (%)	+	0.095
		I2	Annual electricity consumption per capita (KWh)	+	0.052
	Water supply and drainage facilities	I3	Per capita daily consumption of tap water for residential use (L)	+	0.040
		I4	Coverage rate of urban population with access to tap water (%)	+	0.085
		I5	Density of drainage pipeline in developed area (km/km^2^)	+	0.036
		I6	Rate of sewerage disposal (%)	+	0.056
	Transportation facilities	I7	Per captia area of roads (m^2^)	+	0.089
		I8	Number of buses per 10,000 population (unit)	+	0.089
		I9	Number of taxis per 10,000 population (unit)	+	0.057
	Communication facilities	I10	Mobile phone penetration rate (%)	+	0.192
		I11	Internet penetration rate (%)	+	0.049
	Environmental facilities	I12	Per capita park green areas (m^2^)	+	0.043
		I13	Green coverage rate of developed areas (%)	+	0.079
		I14	Public toilets per 10,000 population (unit)	+	0.037
Urban environment	Environmental pollution	E1	Proportion of heavily polluted weather (%)	−	0.203
		E2	Wastewater discharge (10,000 tons)	−	0.074
		E3	SO2, Industrial solid waste and dust (soot) emission volume (t)	−	0.095
		E4	Chemical oxygen demand, NHx and NOx emission volume (t)	−	0.124
	Environmental governance	E5	Per capita water resources (m^3^)	+	0.131
		E6	Green coverage rate of developed areas (%)	+	0.157
		E7	Rate of sewerage disposal (%)	+	0.085
		E8	Comprehensive utilization rate of industrial solid waste (%)	+	0.131

**Table 2 ijerph-19-14909-t002:** Classification of the coupling coordination degree levels.

Index Levels	Coupling	Coordination	Coupling Coordination Types
0.0−0.125	Extreme decoupling	Extreme incoordination	Extreme disorder type
0.125−0.25	Serious decoupling	Serious incoordination	Serious disorder type
0.25−0.375	Moderate decoupling	Moderate incoordination	Moderate disorder type
0.375−0.50	Mild decoupling	Mild incoordination	Mild disease type
0.5−0.625	Primary coupling	Primary coupling	Primary coupling coordination type
0.625−0.75	Moderate coupling	Moderate coordination	Moderate coupling coordination type
0.75−0.875	Favorable coupling	Favorable coordination	Favorable coupling coordination type
0.875−1.0	Quality coupling	Quality coordination	Quality coupling coordination type

**Table 3 ijerph-19-14909-t003:** Selection of the lag order of the model.

Lag	LR	FPE	AIC	SC	HQ
0	NA	3.76 × 10^−9^	−1.676031	−1.542715	−1.63001
1	176.8392 *	2.1 × 10^−9^ *	−6.866236 *	−6.332973 *	−6.682154 *
2	10.96463	2.4 × 10^−9^	−6.743544	−5.810335	−6.4214
3	12.07905	2.56 × 10^−9^	−6.71242	−5.379265	−6.252215

Note: * represents the optimal lag order under this criterion.

## Data Availability

The Microsoft Excel Worksheet data used to support the findings of this study are available from the corresponding author (lishuangchina@cumt.edu.cn) upon request.

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
