# Peer review of "Coupling Coordination and Dynamic Response Analysis of New-Type Urbanization, Urban Infrastructure and Urban Environment—A Case Study of the Jiaodong Economic Circle"

_ijerph, 2022, doi:10.3390/ijerph192214909_

Round 1

Reviewer 1 Report

The research theme of this manuscript is the coupling coordination analysis of new urbanization, urban infrastructure and urban environment. Taking the Jiaozhou Economic Circle as an example, the author establishes a comprehensive indicator system measurement model and a coupling coordination degree model, and empirically analyzes the internal coupling coordination relationship of new urbanization-urban infrastructure-urban environment system. With the acceleration of urbanization, the development of cities has a significant siphon effect on rural areas, resulting in a large number of human, material and financial resources being absorbed into cities, and also leading to the Matthew effect between regions. In the context of new urbanization, China has changed its previous development model and made efforts to build an urban economic circle. Therefore, the topic of this paper has a strong practical significance with the relatively full workload. However, there is still room for improvement in some parts of this manuscript, so I suggest the following major modifications.

1. Introduction

(1) The author shows several over absolute statements. For example, most developing countries have poor urban infrastructure and urban environment. If China wants to explore a sustainable urban development path for developing countries, it is necessary to give play to the synergy among new urbanization, urban infrastructure and urban environment.

(2) In the introduction, in the comparative description of urbanization stage of some countries over the world in 2019, what the reviewers would like to see is that China's urban development has changed from urbanization to new urbanization after so many years. Besides, they would like to figure out whether there is further narrowing, whether there is catching up with some countries, and whether there is still great potential to tap in the future, instead of focusing on existing problems.

2. Research methods

(1) In terms of research methods, the author adopts entropy method and coupling coordination degree model respectively. The former is more objective and the latter is more subjective. Among them, the coupling coordination degree model has problems of subjectivity in indicator construction, volatility and reliability of coupling results. Further correction is able to improve the manuscript at a high level. Reference for Research on mistakes and modification of coupling coordination degree model in China (doi: 10.31497/zrzyxb. 20210319)

(2) As a traditional method, the coupling coordination model cannot solve the endogenous problem. Whether the author considers adopting other models? To do the robustness test can greatly increase the standardization of the article.

3. Logic and Writing

(1) The style of the full manuscript is biased towards the Chinese journal article with the Chinese-style translation. Corresponding corrections are suggested.

(2) The English article generally divides conclusion and discussion into two chapters. In addition, the policy recommendations are too empty, which is unpoweful and inoperable. For example, by considering the vulnerability of urban environment and the increasingly serious urban pollution in the Jijiaozhou Economic Circle, the government should strengthen the construction of urban ecological environment, take effective measures to control the proportion of polluted weather, and figure out what assessment measures should be taken for areas with high population density/severe pollution, areas with low population density/severe pollution, areas with low population density/low pollution, and areas with high population density/light pollution. Policy suggestions can be macro, but they should be targeted and operable meanwhile.

Author Response

Point 1: The author shows several over absolute statements. For example, most developing countries have poor urban infrastructure and urban environment. If China wants to explore a sustainable urban development path for developing countries, it is necessary to give play to the synergy among new urbanization, urban infrastructure and urban environment.

Response 1: Thanks for the reviewer’s question. We have revised some absolute statements in the introduction, as follows:

In China, the most economically dynamic developing country, the process of integrated development of urban agglomerations to compound metropolitan areas has started the influx of population to cities, and after 40 years of reform and opening up, China's urbanization rate has risen from 17.9% in 1978 to 60.6% in 2019 (Fig.1: data from World Development Indicators Database, https://data.worldbank.org.cn). The number of urban population is close to 900 million, which has exceeded the world average urbanization rate by 55%. However, there is still a big gap between the average urbanization rate of developed countries and 80%. China's “Green Paper on Population and Labor: Report No. 22 on China's Population and Labor Issues” predicts that China will see a "turning point" of urbanization from high-speed promotion to gradual slowdown during the "Fourteenth Five Year Plan" period, and enter a relatively stable development stage after 2035. The peak rate of China's urbanization rate is probably 75% to 80%, and China's cities need to receive about 300 million people in the future. In 2019, “The Key Tasks of New-type Urbanization Construction in 2019” released by China shows that the urban population in the future is mainly concentrated in more than ten metropolitan areas, especially the emerging metropolitan areas. Under the trend of new-type urbanization in China, new-type urbanization has put forward new requirements for many carriers carrying urban population. As the material carrier of urban population, urban infrastructure needs to build a more comprehensive urban framework for urban population, and urban environment, as the space carrier of urban population, needs to provide better living space for urban population. Therefore, in the process of transformation from traditional urbanization to new-type urbanization, China's emerging metropolitan area needs to speed up the construction of urban infrastructure and improve the urban environment to accommodate a larger urban population.

Point 2: In the introduction, in the comparative description of urbanization stage of some countries over the world in 2019, what the reviewers would like to see is that China's urban development has changed from urbanization to new urbanization after so many years. Besides, they would like to figure out whether there is further narrowing, whether there is catching up with some countries, and whether there is still great potential to tap in the future, instead of focusing on existing problems.

Response 2: Thanks for the reviewer’s question. We have revised the introduction and mainly discussed China's urban development achievements and future trends, as follows:

In China, the most economically dynamic developing country, the process of integrated development of urban agglomerations to compound metropolitan areas has started the influx of population to cities, and after 40 years of reform and opening up, China's urbanization rate has risen from 17.9% in 1978 to 60.6% in 2019 (Fig.1: data from World Development Indicators Database, https://data.worldbank.org.cn). The number of urban population is close to 900 million, which has exceeded the world average urbanization rate by 55%. However, there is still a big gap between the average urbanization rate of developed countries and 80%. China's “Green Paper on Population and Labor: Report No. 22 on China's Population and Labor Issues” predicts that China will see a "turning point" of urbanization from high-speed promotion to gradual slowdown during the "Fourteenth Five Year Plan" period, and enter a relatively stable development stage after 2035. The peak rate of China's urbanization rate is probably 75% to 80%, and China's cities need to receive about 300 million people in the future. In 2019, “The Key Tasks of New-type Urbanization Construction in 2019” released by China shows that the urban population in the future is mainly concentrated in more than ten metropolitan areas, especially the emerging metropolitan areas. Under the trend of new-type urbanization in China, new-type urbanization has put forward new requirements for many carriers carrying urban population. As the material carrier of urban population, urban infrastructure needs to build a more comprehensive urban framework for urban population, and urban environment, as the space carrier of urban population, needs to provide better living space for urban population. Therefore, in the process of transformation from traditional urbanization to new-type urbanization, China's emerging metropolitan area needs to speed up the construction of urban infrastructure and improve the urban environment to accommodate a larger urban population.

Point 3: In terms of research methods, the author adopts entropy method and coupling coordination degree model respectively. The former is more objective and the latter is more subjective. Among them, the coupling coordination degree model has problems of subjectivity in indicator construction, volatility and reliability of coupling results. Further correction is able to improve the manuscript at a high level. Reference for Research on mistakes and modification of coupling coordination degree model in China (doi: 10.31497/zrzyxb. 20210319)  

Response 3: Thanks for the reviewer’s question. We have revised the coupling coordination model by referring to the paper you mentioned and other papers, as shown in:

The coupling degree  is the core part of CCDM, and the value of  should be in  to indicate the strength of the coupling relationship between systems. The wrong use of CCDM mainly comes from the wrong coupling degree  formula, which leads to the  result interval not being , thus causing a series of errors in the results and interpretation of the subsequent coupling coordination degree. The CCDM of multiple subsystems are mainly divided into two categories. The first category is the equation (6) used by Ma [49] and Liu [50] to calculate the  value. Jiang [51] proved that the value range of  calculated by equation (6) is , so equation (6) will underestimate the coupling degree, so the conclusion obtained by using equation (6) to calculate the  value is not valid. The second category is equation (7) used by Ge and other scholars [52] to calculate the value. Jiang [51] and Wang [53] proved that the value range of coupling degree  in equation (6) is between , when and only when , the coupling degree reaches the maximum value of 1. Equation (7) is correct.

..............................

Point 4: As a traditional method, the coupling coordination model cannot solve the endogenous problem. Whether the author considers adopting other models? To do the robustness test can greatly increase the standardization of the article.

Response 4: Thanks for the reviewer’s question. We added PVAR model to study the dynamic response relationship of NTU, UI and UE. PVAR model can well solve the endogenous problem of variables. We also tested the robustness of PVAR model with three variables and successfully passed the robustness test. The specific modification contents are shown in Sections 3.6 and 4.3 of the manuscript.

Point 5: The style of the full manuscript is biased towards the Chinese journal article with the Chinese-style translation. Corresponding corrections are suggested.

Response 5: Thanks for the reviewer’s question. We have revised the overall content of the manuscript.

Point 6:  The English article generally divides conclusion and discussion into two chapters. In addition, the policy recommendations are too empty, which is unpoweful and inoperable. For example, by considering the vulnerability of urban environment and the increasingly serious urban pollution in the Jijiaozhou Economic Circle, the government should strengthen the construction of urban ecological environment, take effective measures to control the proportion of polluted weather, and figure out what assessment measures should be taken for areas with high population density/severe pollution, areas with low population density/severe pollution, areas with low population density/low pollution, and areas with high population density/light pollution. Policy suggestions can be macro, but they should be targeted and operable meanwhile.

Response 6: Thanks for the reviewer’s question. We have revised the conclusions and recommendations as follows:

  1. Conclusion and implications for urban integration

Based on the panel data of five cities in Jiaodong Economic Circle from 2010 to 2019, this paper establishes EWM-CCDM-PVAR model to analyze the development level, coupling coordination degree and dynamic response relationship of NTU-UI-UE in Jiaodong Economic Circle. The results show that: (1) The development level of new-type urbanization and urban infrastructure in Jiaodong Economic Circle has been steadily improved, but there are regional differences in the development of new-type urbanization and urban infrastructure in Jiaodong Economic Circle, and the differences are expanding year by year. The overall development is uneven. The development of urban environment fluctuates greatly, and the urban environment of most cities in Jiaodong Economic Circle is deteriorating. (2) The CCD level of NTU-UI-UE in Jiaodong Economic Circle is on the rise, but the CCD level of NTU-UI-UE in Weifang and Yantai is only Moderate coupling coordination type, and the development within the region is uneven. (3) NTU and UI of Jiaodong Economic Circle have established a mutually reinforcing dynamic response relationship, and UI and UE have also established a mutually reinforcing dynamic response relationship, but NTU and UE have a mutually inhibiting dynamic response relationship. (4) During the ten-year development process of establishing a metropolitan area by retaining the independent administrative divisions of cities and achieving interregional urban integration in the Jiaodong Economic Circle, the development levels of new-type urbanization, urban infrastructure, and the coupled coordination degree level of the NTU-UI-UE system have been substantially improved, the regional integration continues to deepen. Under the coordinated governance and mutual achievements of new-type urbanization, urban infrastructure and urban environment, Jiaodong Economic Circle has formed a metropolitan circle pattern with “one core” and “four wings”, which confirm that the urban development path with Chinese characteristics is feasible and correct.

We put forward the following suggestions for the urban development of the new economic circle: First, the urbanization process will bring environmental pollution, and the poor urban environment will also hinder the urbanization process.Different environmental governance measures should be implemented according to different industrial structures in different regions. There are many heavy industries in Jiaodong Economic Circle. Environmental policies and measures aimed at strengthening the end treatment and clean transformation of highly polluting industries, and improving energy efficiency can help Jiaodong Economic Circle quickly surpass the peak of pollutants. Second, in the process of urban integration in the emerging economic circle, regional differences are the main problem. The industrial structure is the fundamental problem that hinders urban integration. In the process of the integration of the emerging metropolitan area, it is necessary to promote the industrial complementation of the cities within the metropolitan area. World class metropolitan areas have formed distinctive industrial clusters [60]. Drawing on the experience of world class metropolitan areas, Jiaodong Economic Circle should coordinate the regional industrial division and cooperation system. Weifang and Yantai have obvious advantages in manufacturing industry, Yantai and Rizhao have obvious advantages in tourism service industry, Qingdao has developed finance, trade and logistics, the five cities have the basis of industrial complementation, and the prospect of industrial cooperation is great. Higher level industrial cooperation will make up for regional differences.

Reviewer 2 Report

The paper is interesting as an idea. However, it needs serious extensions and revisions and there are many points for authors to address. But the authors did not explain why it is interesting to have a look at this topic and what could be done based on the results.

The introduction is quite verbose and is a combination of a literature review. It does not cover the motivation and the contribution of the work. It needs proper extensions.

The explanation of the entropy weight method needs to be improved to give the reader a better understanding how it works and selected to evaluate the system.

The manuscripts research method is not that innovative.

Besides, the map of China should be downloaded according to the standard map.

Author Response

Point 1: The introduction is quite verbose and is a combination of a literature review. It does not cover the motivation and the contribution of the work. It needs proper extensions.

Response 1: Thanks for the reviewer’s question. We have appropriately deleted the introduction and added the purpose and contribution of the article, as follows:

The development process of cities in the world can be seen as the combination of the great growth of the number of cities, the great development of urban circles and the increasingly formed urban agglomerations, as well as the increasingly mature coordinated development pattern of large, medium and small cities and towns, thus forming a metropolitan circle and moving towards a comprehensive convergence among cities [1]. In the process, countries in the world have built metropolitan areas by merging and adjusting large-scale cities in recent centuries [2,3]. The basic meaning of urban consolidation refers to the process that urban municipal districts realize scale expansion by absorbing or combining other regions to form a single administrative region [4]. Overall, there are three main types of city consolidation, namely Annexation, Consolidation and Merger [5]. Annexation refers to the incorporation of unincorporated areas or small towns into the administrative boundaries of a city, and is widely used in developed countries; Consolidation refers to the merger of different levels of administrative districts, which is more prominent in the United States, mainly in the form of city-county consolidation [6]; Merger refers to the merger of administrative districts at the same level, which is more popular in European countries, mainly between two or more cities [7,8,9,10,11]. Despite the different forms, developed countries as a whole are showing a trend of urban expansion and shaping a new spatial pattern of “great state and bigger city”. Developed countries have largely completed the urbanization process and established world-class metropolitan areas, developing countries are entering a period of rapid urbanization, with the BRICS countries of China, Brazil, Russia and South Africa topping the list of urbanization rates and even surpassing some developed countries [12]. In its urban development process, China, the largest developing country, did not follow the process of urban development in developed countries by engaging in large-scale urban mergers, but rather retaining the separate administrative divisions of cities, while achieving inter-regional urban integration, followed by the establishment of metropolitan areas, with a view to providing the developing world with a sustainable urban development path.

Industrial agglomeration and population concentration are the two twin conditions of urbanization, especially the concentration of urban population has become an important symbol of the development process of urbanization in the world, and is also a necessary condition for the rapid promotion of urbanization and even the formation of metropolitan areas [13]. Globally, the current general urbanization level is about 55%, and the United Nations predicts that nearly 1.1 billion new urban residents will be absorbed by cities in 2015-2030 [14]. In China, the most economically dynamic developing country, the process of integrated development of urban agglomerations to compound metropolitan areas has started the influx of population to cities, and after 40 years of reform and opening up, China's urbanization rate has risen from 17.9% in 1978 to 60.6% in 2019 (Fig.1: data from World Development Indicators Database, https://data.worldbank.org.cn). The number of urban population is close to 900 million, which has exceeded the world average urbanization rate by 55%. However, there is still a big gap between the average urbanization rate of developed countries and 80%. China's “Green Paper on Population and Labor: Report No. 22 on China's Population and Labor Issues” predicts that China will see a "turning point" of urbanization from high-speed promotion to gradual slowdown during the "Fourteenth Five Year Plan" period, and enter a relatively stable development stage after 2035. The peak rate of China's urbanization rate is probably 75% to 80%, and China's cities need to receive about 300 million people in the future. In 2019, “The Key Tasks of New-type Urbanization Construction in 2019” released by China shows that the urban population in the future is mainly concentrated in more than ten metropolitan areas, especially the emerging metropolitan areas. Under the trend of new-type urbanization in China, new-type urbanization has put forward new requirements for many carriers carrying urban population. As the material carrier of urban population, urban infrastructure needs to build a more comprehensive urban framework for urban population, and urban environment, as the space carrier of urban population, needs to provide better living space for urban population. Therefore, in the process of transformation from traditional urbanization to new-type urbanization, China's emerging metropolitan area needs to speed up the construction of urban infrastructure and improve the urban environment to accommodate a larger urban population.

Figure 1. The stage of urbanization in some countries of the world in 2019

The urban system (NTU-UI-UE) composed of new-type urbanization (NTU), urban infrastructure (UI) and urban environment (UE) has been considered as an important basis for urban population expansion by the academic community. In terms of research themes and contents, there are more studies around the interaction between new-type urbanization, urban infrastructure and urban environment, which mainly involve the establishment of two-dimensional evaluation systems: e.g. new-type urbanization-environment [15], new-type urbanization-infrastructure [16], urbanization-land use [17], economy-environment [18], infrastructure-environment [19], etc. In the research method, the coupling coordination degree (CCD) model [12], dynamic simulation model [17], spatial association model [20] and other methods were used for empirical analysis, or the development relationship between two-dimensional systems was analyzed qualitatively in theory [21,22]. From the perspective of research area, the current research involved the national [23], province [24] and prefecture level cities [25]. At the national level, the study of NTU-UI-UE coupling coordination relationship at the provincial scale is large, which easily lead to ignore the spatial and temporal variability of NTU-UI-UE within the province, while the studies at the scale of prefecture-level city mostly focus on the development within the municipal area, ignoring the integration process between regions. Thus, it is essential to include urban agglomerations in the coupling coordination study of NTU-UI-UE, as most of the urban agglomerations studies are distributed in the more mature metropolitan areas.

Based on the above research, there are mainly the following deficiencies: (1) The research scope is insufficient. the above researches adhered to the two-dimensional discussion of new-type urbanization and urban environment, urban infrastructure and population urbanization, new-type urbanization and single infrastructure system, and the multi-dimensional research on new-type urbanization, urban infrastructure and urban environment was lacking. (2) In terms of study area, the study area is mainly national, provincial, prefecture level cities and mature metropolitan areas, while the future urban population is mainly planned in emerging metropolitan areas, so it is urgent to study emerging metropolitan areas. (3) In terms of research methods, on the one hand, they were to qualitatively describe the connotation relationship among new-type urbanization, urban infrastructure and urban environment. On the other hand, they were to quantitatively analyze the coordination relationship among new-type urbanization and urban infrastructure, urban environment based on statistical data, which lacks a combination of qualitative and quantitative research methods. Qualitative description is difficult to objectively understand the development trend of new-type urbanization, urban infrastructure and urban environment. Empirical analysis can only explore the interaction between the three, and cannot clarify the coordination degree of the three.

In order to make up for the shortcomings of previous research, this paper takes Jiaodong Economic Circle as the research area, constructs NTU-UI-UE, establishes three indicator systems of new-type urbanization, urban infrastructure and urban environment, and measures the development level of new-type urbanization, urban infrastructure and urban environment through the entropy weight method (EWM). NTU-UI-UE forms an organic whole of a city with complex content structure and complex interaction coupling relationship, coupling coordination degree model (CCDM) is used to discuss the coupling coordination relationship between new-type urbanization, urban infrastructure and urban environment, and PVAR model is used to jointly explain the dynamic response relationship between new-type urbanization, urban infrastructure and urban environment in a qualitative and quantitative way.

In a word, this paper established EWM-CCDM-PVAR model to evaluate the development level, coupling coordination degree and dynamic response relationship of urban NTU-UI-UE, clarify the problems and development direction of China's emerging metropolitan area in the process of building a framework to support urban population, and provide reference for China's urbanization process, especially in developing countries.

Point 2: The explanation of the entropy weight method needs to be improved to give the reader a better understanding how it works and selected to evaluate the system.

Response 2: We have reinterpreted the reasons for solving the problem with the method of selection entropy weight, as follows:

NTU-UI-UE is a huge complex system, which has the characteristics of element set, clear hierarchy, strong correlation and internal nonlinear effect. According to the dissipation theory, the internal positive entropy flow is the main cause of system disorder in the process of system evolution. With the increase of positive entropy, the degree of internal system disorder is deepened [48].

In the evolution process of NTU-UI-UE, the interaction of internal factors of the system as the internal positive entropy flow is the main reason for the disordered development of the system. Therefore, it is necessary to clarify the internal problems of the system, provide materials for the internal system through external urban planning, and promote information exchange. Therefore, to determine the weight of NTU-UI-UE indicators, the entropy weight method based on the degree of chaos (entropy) within the system should be selected as the evaluation method.

Entropy is an important concept in the second law of thermodynamics to characterize the state of matter, which was originally introduced to information theory by Shannon [48] to represent the degree of uncertainty, and is now commonly used in the field of urban development [36].

Point 3: The manuscript’s research method is not that innovative.

Response 3: Our research method is applicable to our research problems. We deleted the obstacle model and added the PVAR model to explore the dynamic response relationship between NTU-UI-UE. See Sections 3.6 and 4.3 in the manuscript for details.

Point 4: Besides, the map of China should be downloaded according to the standard map.

Response 4: Thanks for the reviewer’s question. We used Arcgis to modify the map of Jiaodong Economic Circle, as follows:

Figure 2. Study area

Round 2

Reviewer 2 Report

Now the research is fine to me.